# Food Confusion Between Edible and Poisonous Plants: A 22-Year Retrospective of the Southeastern France Poison Control Center

**DOI:** 10.3390/toxins16120552

**Published:** 2024-12-21

**Authors:** Romain Torrents, Julien Reynoard, Mathieu Glaizal, Corinne Schmitt, Katharina Von Fabeck, Audrey Boulamery, Luc De Haro, Nicolas Simon

**Affiliations:** 1Aix Marseille Univ, APHM, INSERM, IRD, SESSTIM, Hôpital Sainte Marguerite, Clinical Pharmacology and Poison Control Centre, 13274 Marseille, France; nicolas.simon@ap-hm.fr; 2APHM, Hôpital Sainte Marguerite, Clinical Pharmacology and Poison Control Centre, 13274 Marseille, France; julien.reynoard@ap-hm.fr (J.R.); mathieu.glaizal@ap-hm.fr (M.G.); corinne.schmitt@ap-hm.fr (C.S.); moenikes@ap-hm.fr (K.V.F.); audrey.boulamery@ap-hm.fr (A.B.); luc.deharo@ap-hm.fr (L.D.H.)

**Keywords:** food confusion, botanical intoxication, accidental ingestion

## Abstract

Objective: In some regions of the globe, accidental food confusion regarding plants can cause severe poisoning events and deaths. The aim of this study was to report on those confusions from the Marseille Poison Control Centre’s (PCC) experience from 2002 to 2023. Results: Over 22 years, 2197 food confusion events were managed with 321 different species. The most frequently involved plant was Nerium oleander (289 cases, 13.1%), then Cucurbitaceae genus (3.3%), Colchicum autumnale (3.3%), Prunus amygdalus (3%), Mahinot esculenta (3%), Cytisus laburnum (2.6%), Aesculus hippocastanum (2.5%) and Narcissus Jonquilla (2%). Many botanical confusion events were also reported (*n* = 1386, 63%), but with fewer than five identical species. Only one death was reported for this review, in an event involving Aconitum napellus. Two antidotes were used for all the series: Datura genus and Prunus dulcis. Discussion: The most implicated plant was Nerium oleander. This is explained by its distribution. This rate is very low compared to that of suicide attempts with this plant. Many cases were symptomatic (53.6%), but very few of them described severe symptoms (only 0.5% severe poisonings). Few patients needed to be admitted to intensive care (0.4%), and even fewer needed an antidote (two cases). Only one death occurred, involving Aconitum napellus. Food confusions were more common than suicide attempts with plants, but seem to be less severe. However, death and serious complications can occur, so it is important to identify and manage the plants concerned. Material and Methods: For each food confusion event managed between 2002 and 2023 at the Southeastern France PCC based in Marseille, we performed a retrospective review. This PCC is responsible for the Provence–Alpes–Côte d’Azur Region, Corsica Island and Indian Ocean French overseas territories. For each case, severity was calculated with the Poison Severity Score (PSS).

## 1. Introduction

Human intoxication via plants is common, but the rate of toxicity and fatality is low [1]. Deaths are even more rare in the industrialized world, especially in France. However, botanical intoxication is a significant medical issue in some areas of the developing world. Most cases require symptomatic treatment, and only a few need an antidote.

Three plant poisoning causes can be described: misuse, deliberate self-poisoning and accidental [1]. Many misuse-type plant exposition events have been described [2,3]. In a previous work, we studied deliberate self-poisoning events over 20 years to evaluate their clinical severity and the incidence of specific treatment [4]. We are now interested in studying accidental ingestion during the same period, excluding unintentional poisoning in young children. Indeed, that is the most common circumstance; they often put anything they want to identify to their mouth, including plants [5,6], and so there are many descriptions of cases involving children [7,8,9,10,11]. Confusion between toxic and edible plants can lead to severe poisoning [12], but this is less frequently described. Many descriptions are case reports or smaller case series [13,14,15,16,17].

Therefore, with this retrospective study, we wanted to estimate plant food confusion event characteristics managed by the Southeastern PCC and to determine which plants can induce high rates of toxicity and fatality.

The study aim was to describe those confusions and to evaluate their clinical severity and which required specific treatment for human ingestion in Southeastern France over 22 years.

## 2. Results

According to Table 1, 2197 cases of plant food confusion were reported among 513,750 total reports to the PCC during those 22 years. The most frequently involved species in food confusion events was Nerium oleander, responsible for 289 poisonings (13.1%). Then, we report 73 intoxications with the Cucurbitaceae genus (3.3%), and 72 confusions with Colchicum autumnale (3.3%). Many other confusions were reported with Prunus amygdalus (3%), Mahinot esculenta (3%), Cytisus laburnum (2.6%), Aesculus hippocastanum (2.5%) and Narcissus Jonquilla (2%). There were 316 other plants consumed in this series, but with fewer than five poisonings each. These 316 species constituted 63.1% of the whole study. Referring to the PSS, eleven food confusion events were severe, with only one due to Nerium oleander, two to Colchicum autumnale, one to Cytisus laburnum, two to Datura genus and two to Aconitum napellus. The two other severe poisonings were due to Hippomane mancinella and Conium maculatum. Only one death was reported, due to Aconit Napellum. Among all 289 incidences of Nerium oleander consumption, no Digifab was used. Only two antidotes were utilized during the whole study: Hydroxocobalamin for a Prunus amygdalus poisoning and physostigmine for a Datura genus poisoning.

Table 2 reports the mean age: 34 ± 26 years. Women were principally involved in plant food confusion (60.6%). Many patients were symptomatic (N = 1178), although there were 627 with vomiting or nausea (28.5%), 269 with abdominal pain (12.2%) and 193 with diarrhea (8.8%). Also, 31 patients (1.4%) had tachycardia, 22 reported bradycardia (1%), 22 had low blood pressure (1%), 10 reported drowsiness (0.4%), 6 were in coma (0.3%) and 6 had seizures (0.3%). Many hospitalizations were reported (N = 190, 8.6%), but few in an intensive care department (0.4%). According to PSS, 0.5% patients were classified as having severe intoxication. Some parameters were statistically significant between the 289 *Nerium oleander* confusions and the 1898 other confusions. Mostly women were involved (60.6%), but significantly less with *Nerium oleander* than with other plants (*p* = 0.01). Fewer patients who consumed *Nerium oleander* reported symptoms compared to after consuming other plants (*p* < 0.01), with less nausea/vomiting (*p* < 0.01), diarrhea (*p* < 0.01) and abdominal pain (*p* < 0.01) experienced, but with more bradycardia (*p* = 0.04). No statistical differences were shown for the other parameters described.

## 3. Discussion

The most common plant food confusion involved *Nerium oleander* (13.1% of the study), which can easily be confused with the edible *Laurus nobilis*. This can be imputable to the high distribution of this local autochthonous plant in the areas managed by the Marseille PCC. That is why we chose to compare this plant’s poisoning events versus all the other ones, like we did in our previous study concerning suicides attempts [4]. *Nerium oleander* confusions seemed to concern fewer women, with fewer digestive symptoms but with more bradycardia than confusions with other plants. Moreover, this 13.1% rate of *Nerium oleander* food confusion was very low compared to the previous rate of 71% that we described in our previous study concerning suicide attempts with this plant [4]. This can be explained by the diversity of the species ingested by confusion compared to the previous study concerning the suicide attempts (331 species versus 43) [4]. Among those 289 *Nerium oleander* poisonings in this study, only 1 severe poisoning event was described, with severe bradycardia involving atropine use, but anti-digoxin antibodies were not necessary. Usually, an important Fab antibody posology is necessary to recovery without sequalae. Some cases even needed repeated administrations. But this specific treatment is very costly, and availability can be very challenging [4]. Digestive disorders (emesis, nausea and epigastralgia) are common with this species [18]. The cardiac symptoms are caused by a positive inotropic effect due to the inhibition of cardiomyocyte Na^+^/K^+^ ATPase, the same way we can see with digitalis glycosides [18].

The second most incriminated species belong to the Cucurbitaceae genus. Digestive symptoms can occur after eating non-edible squash mistaken for similar looking species [19]. This can lead to a bitter taste and a drastic purgative action, as appears in our report, but we described no cases of death or severe poisoning. The digestive toxicity of this family is linked to the presence of bitter and cytotoxic molecules called cucurbitacins, but the correlation between cucurbitacin and bitterness/toxicity is still unknown [19].

*Colchicum autumnale* can easily be confused with the edible *Allium ursinum* [16]. We reported two severe cases with *Colchicum autumnale*. The clinical manifestations of colchicine poisoning are classically present in three phases. Vomiting and diarrhea characterize the first phase. The second phase is characterized by severe aggravation, such as acute cardiac, respiratory, kidney and hepatic failures, rhabdo-myolysis and bone marrow suppression [16]. After one week, if patients survive the second phase, it is followed by a third phase, with neuropathy and alopecia [20]. Colchicine-specific anti-bodies are unavailable, so therapy is only symptomatic [21]. Bone marrow suppression with thrombocytopenia and leucopenia developed in the two severe poisoning cases in our study.

The ingestion of *Cytisus laburnum* usually only causes gastrointestinal upset [22]. Serious cases of poisoning are an exception. Among our 57 poisoning events with this plant, only one was severe. This patient, who believed that *Cytisus laburnum* was edible, developed a cardiovascular collapse a few hours after taking an infusion, leading to adrenalin use in the emergency department. Fatal cases of poisoning are extremely rare due to the compound’s innate emetic effect, but have already been described [22].

*Datura genus* is one of the most well known and most abused psychoactive plants [23]. But it can also be confused with edible plants like *Spinacia oleracea* [24]. It induces anticholinergic poisoning, as it includes atropine, scopolamine and hyoscyamine [23]. The typical clinical signs of intoxication are dryness of skin and mucous membranes, reddening of the skin, hyperthermia, tachycardia, visual defects, constipation, urinary retention, hallucination, disorientation and agitation. In our study, two severe cases of poisoning were reported with *Datura genus*, one of which needed the use of physostigmine to improve the neurological symptoms.

No severe poisoning cases were reported with *Ricinus communis* in this study. This is very surprising because ricin is considered to be one of the most potent poisons known [25]. It is possible that this was because ricin is only released if the seed is crushed, and in most of our confusion events this was not the case.

The only death reported in this study was with the ingestion of *Aconitum napellus*, confused with the edible *Molopospermum peloponnesiacum* [26]. *Aconitum napellus* is known to be very dangerous, and other severe poisoning events have been described with this plant [27,28]. We can say that is the most dangerous plant in our study, because among only six confusion events with it, there was one death and two severe poisonings reported. Aconite roots contain aconitine, mesaconitine, hypaconitine and other Aconitum alkaloids, which are known cardiotoxins and neurotoxins and can induce life-threatening ventricular arrhythmias [27]. Our patient died of cardiovascular decompensation on acute arrhythmia despite supportive management in intensive care.

Some plant material involves natural amygdalin and, after ingestion, this produces cyanide through the hydroxylation of glucosidase and emulsion enzymes [Bıcılıoğlu].

For mild cases, supportive treatment is sufficient; on the other hand, antidote treatment must always be given for cases with mood changes, severe metabolic acidosis, resistant convulsion and hypotension. A cyanide antidote kit Hydroxocobalamin (Cyanokit) can be used as an antidote [29]. In our study, one case of *Prunus amygdalus* ingestion involved severe metabolic acidosis, resistant convulsion and hypotension, which required this antidote for recovery without sequelae.

The other plants ingested, such as *Manihot esculenta, Aesculus hippocastanum*, *Narcissus jonquilla*, *Jatropha curcas*, *Myristica fragrans*, *Sambucus nigra* and *Taxus baccata*, caused no severe symptoms in this study. Only digestive symptoms occurred, such as nausea, abdominal pain, vomiting and diarrhea. However, severe poisonings have previously been described to occur with *Myristica fragrans* [30] and *Taxus Baccata* [31].

Those results seem to complement the confusion events registered by the other French poison control centres [12]. Indeed, there was the same rate of symptomatic patients (53.5%). The most incriminated species were quite similar—bitter squash, cytisus, horse chestnut—but the most frequent plants reported were bulbs (narcissus, daffodil, crocus, etc.) and not *Nerium oleander* [12]. Severe poisonings were also very rare and only concerned 14 cases.

Food confusion events were far more common than suicide attempts with plants (2197 cases in 22 years versus 262), but seemed to be less severe (0.5% of severe cases in this series versus 16% with suicide attempts) [4]. One hypothesis could be that quantity is less important in food confusion than in an intentional context, where the amount of plants ingested can be massive. Unfortunately, we cannot be sure of the exact amount of toxic plants consumed in our study because it is rarely precisely described.

## 4. Conclusions

The most common plant food confusion events involved *Nerium oleander*. This can be imputable to the high distribution of this local autochthonous plant in the areas managed by the Marseille PCC. Many cases were symptomatic (53.6%), and very few of them described severe symptoms (only 0.5% severe poisonings). Few needed to be admitted to intensive care (0.4%), and even fewer required an antidote (two cases). Only one death occurred, involving *Aconitum napellus.* Food confusions were more common than suicide attempts with plants, but seem to be less severe. However, death and serious complications can occur, so it is important to identify and manage them. Public knowledge is also required to avoid those dangerous confusions.

## 5. Materials and Methods

The methodology of this study has already been detailed in a previous study concerning suicide attempts (data source, file selection, data collected and statistics) [4]. In order to compare those two studies, we tried to use exactly the same method. The only difference is the circumstances of poisoning: food confusions versus suicide attempt. We also added data from the two years, 2022 and 2023, between the two studies.

Data were collected by the Marseille PCC, which explores all files reporting botanical exposition and self-poisonings, from 1 January 2002 to 31 December 2023.

A blinded review of each file was completed twice by two medical toxicologists.

For each food confusion event managed between 2002 and 2023 at the Southeastern France PCC based in Marseille, we performed a retrospective review. This PCC is responsible for the Provence–Alpes–Côte d’Azur Region, Corsica Island and Indian Ocean French overseas territories (Reunion Island, Mayotte).

Severity was evaluated using the Poison Severity Score (PSS) [32].

Severity grades were classified as none (0), minor (1), moderate (2), severe (3) and fatal (4).

Student’s *t*-test was used for continuous variable comparison and the chi-squared test or Fisher’s exact test was used for qualitative variable comparison.

## Figures and Tables

**Table 1 toxins-16-00552-t001:** Food confusion events with plants managed by the Marseille PCC between January 2002 and December 2023.

Species Ingested	Patients (%)	Patients with Symptoms	Severe Intoxication	Death	Specific Treatment (Antidote)
*Nerium oleander*	289 (13.1%)	64	1	0	0
*Cucurbitaceae genus*	73 (3.3%)	36	0	0	0
*Colchicum autumnale*	72 (3.3%)	41	2	0	0
*Prunus amygdalus*	67 (3%)	20	1	0	1
*Mahinot esculenta*	67 (3%)	12	0	0	0
*Cytisus laburnum*	57 (2.6%)	44	1	0	0
*Aesculus hippocastanum*	56 (2.5%)	23	0	0	0
*Narcissus jonquilla*	44 (2%)	34	0	0	0
*Jatropha curcas*	34 (1.5%)	18	0	0	0
*Datura genus*	24 (1.1%)	19	2	0	1
*Myristica fragrans*	15 (0.7%)	11	0	0	0
*Sambucus nigra*	15 (0.7%)	13	0	0	0
*Ricinus communis*	14 (0.6%)	8	0	0	0
*Taxus baccata*	7 (0.3%)	1	0	0	0
*Aconitum napellus*	6 (0.3%)	4	2	1	0
Other plants (max. 5 cases each)	1386 (63.1%)	830	2	0	0
Total	2197	1178	11	1	2

**Table 2 toxins-16-00552-t002:** *Nerium oleander* food confusion events versus other plants managed by the Marseille PCC between January 2002 and December 2023.

	Plant Confusion (N = 2197)	*Nerium oleander* Confusion (N = 289)	Other Species Confusions (N = 1898)	*p*-Value
Age (Mean +/− SD)	34 +/− 26	38 +/− 22	39 +/− 23	0.5
Female	1331(60.6%)	151 (52.5%)	1180 (61.8%)	0.01
Symptomatic patients	1178 (53.6%)	64 (22.1%)	1114 (58.4%)	<0.01
Nausea/vomiting	627 (28.5%)	30 (10.4%)	597 (31.4%)	<0.01
Abdominal Pain	269 (12.2%)	8 (2.8%)	261 (13.7%)	<0.01
Diarrhea	193 (8.8%)	11 (3.8%)	182 (9.6%)	<0.01
Tachycardia	31 (1.4%)	1 (0.3%)	30 (1.6%)	0.07
Bradycardia	22 (1%)	6 (2.1%)	16 (0.8%)	0.04
Low blood pressure	22 (1%)	0 (0%)	22 (1.2)	0.1
Drowsiness	10 (0.4%)	1 (0.3%)	9 (0.5%)	0.4
Coma	6 (0.3%)	0 (0%)	6 (0.3%)	0.4
Seizures	6 (0.3%)	0 (0%)	6 (0.3%)	0.4
Hospitalization	190 (8.6%)	16 (5.5%)	174 (7.9%)	0.06
Intensive care	9 (0.4%)	1 (0.3%)	8 (0.4%)	0.09
Severe Poisonings (PSS = 3)	11 (0.5%)	1 (0.3%)	10 (0.5%)	0.07

## Data Availability

The original contributions presented in this study are included in the article. Further inquiries can be directed to the corresponding author.

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
