# Peer review of "Food Confusion Between Edible and Poisonous Plants: A 22-Year Retrospective of the Southeastern France Poison Control Center"

_toxins, 2024, doi:10.3390/toxins16120552_

Round 1

Reviewer 1 Report

Comments and Suggestions for Authors

In this manuscript, the authors carry out a retrospective study to estimate accidental poisoning by toxic plants mistaken for food based on the experience of the Marseille Poison Control Center (CCP) from 2002 to 2023. Although the topic addressed in this article is worthy of attention, the manuscript is too descriptive, simple and short and in my opinion does not have the entity to be published as an article and would fit better as a short communication.

The article needs to improve notably in many aspects. The introduction is too short, and the results should be described in more detail.  

For example, it would be interesting to mention some of the toxic plants consumed by 830 patients with symptoms and to know which plants (included in other plants) caused severe intoxication to 3 patients and which antidote was used in one of those cases.

In the discussion, it would be interesting to comment on all the ingested species mentioned in table 1. Nothing is commented about Prunus amygdalus, Manihot esculenta, Aesculus hippocastanum, Narcissus jonquilla, Jatropha curcas, Myristica fragrans, Sambucus nigra and Taxus baccata.

On the other hand, it would be interesting to explain why is relevant to compare the symptoms of patients who consumed Nerium oleander with those with other confusions.

It would also be interesting to comment on the amount of toxic plants consumed to cause those effects.

Minor comments:

-Line 10 . “The first involved plant was Nerium oleander (283 cases, 13.1%)”. According to data in table 1 there were 289 cases.

-Table 1.Prunus amygdalus and Prunus dulcis are synonymous. Line 59: ”Methylene blue was used for a Prunus dulcis poisoning”. However, in the specific treatment (Antidote) column for Prunus amygdalus, the number 1 does not appear.

-Line 86. “This can be explained by the diversity of the species ingested by confusion compared to the previous study concerning the suicide attempts (331 species versus 43) [4]”. Confirm that those numbers are correct.  Could be 321 species versus 35 or 37?

-Line 109.  Change [Musshoff] by reference 22

-Line 116.  Change “But is can also be confused….”  by “But it can also be confused….” .

Author Response

please Check attached files below

Reviewer 2 Report

Comments and Suggestions for Authors

This retrospective study describes accidental poisonings with plants that were reported to a poison center in France during the last 22 years. Although retrospective, the study is of interest since it presents current data that could be used to raise awareness among the public and thus contribute to public health. However, some adjustments would improve the quality of the paper.

Some specific comments and suggestions:

-       The manuscript would benefit from language editing by a native speaker. It might also be of advantage to use the term “accidental poisoning with plants” instead of e.g. “food confusion” and use the past tense throughout the manuscript (e.g. “were” instead of “are”).

-       Abstract, discussion section: The numbers regarding Nerium oleander (13.1%) are a repetition from the results; the 71% reported on the other hand is not from the current study, thus probably better not to mention it in the abstract

-       Introduction: The sentence “The study was conducted between January 2002 and December 2023” would be better placed in the methods section

-       Results, “…2197 cases of plant food confusion were reported”; out of how many total reports to the poison center during the same time period?

-       Results, “There were 316 other plants…with less than 5 poisonings each”: Are they reported somewhere currently? Could be added as e.g. an appendix if too long

-       Results, “…11 food confusions were severe, only one due to Nerium oleander”: Could you add the information which plants were involved in the other 10 severe cases?

-       Table 2 and text results section: To be added that age is presented as mean +/- SD (currently only mean +/-)

-       Results, “According to PSS, 0.5% patients described severe intoxication”: I think the PSS is based also on lab and clinical findings, not only the description by the patient (who might not be able to give any information if e.g. unconscious)

-       In the results you currently describe some comparisons that were performed, it should be added in the methods what you compared and how/which tests you used

-       Results, “No statistical differences were shown for the other parameters described”: According to Table 2 there were some more significant differences, e.g. for gender

-       Table 2: Was gender or sex assessed?

-       Discussion: You could also add how your findings are compared with other similar studies (e.g. some of the references you mention in the introduction)

-       Discussion, “This can be explained by the diversity of the species ingested”: Pethaps also due to the often greater amounts ingested in case of a suicide attempt compared to accidental poisoning?

-       In the discussion some information regarding the available evidence for some of the therapies mentioned (e.g. anti-digoxin antibodies for Nerium oleander) could be added

-       It would also be very interesting and also important from an educational point of view if you could show some pictures of the reported poisonous plants next to the harmless similar ones they were confused for

-       Materials and Methods, “The methodology of this study has already been detailed in a previous study…”: It would be helpful if you could add some very short information regardless about the main points of the methods, e.g. was the search text- or ICD-based, search terms used, how many people reviewed each case etc.

-       Materials and Methods, “Severity was evaluated… (PSS)”: You could shortly mention the categories of the PSS (except from “severe”)

Round 2

Reviewer 1 Report

Comments and Suggestions for Authors

In this revised version of the manuscript, the authors have addressed most of the reviewer’s comments and improved the paper. In the current version the content and tables are better explained.

Author Response

Thank you for the review

Reviewer 2 Report

Comments and Suggestions for Authors

Thank you for addressing all the comments. One minor correction: “File selection: All files reporting botanical exposition and self-poisonings, from 1 January 2002 to 31 December 31 2021” -> I think this should be 31 December 2023.

Author Response

Thank you for the review.

Your correction has been modified.